# Managed Retreat as Adaptation Option: Investigating Different Resettlement Approaches and Their Impacts—Lessons from Metro Manila

**Hannes Lauer** [1,*] , **Mario Delos Reyes** [2] and **Joern Birkmann** [1]

1 Institute of Spatial and Regional Planning (IREUS), University of Stuttgart, 70049 Stuttgart, Germany; joern.birkmann@ireus.uni-stuttgart.de

2 School of Urban and Regional Planning (SURP), University of the Philippines, Quezon City 1101, Philippines; mrdelosreyes@up.edu.ph

* Correspondence: hannes.lauer@ireus.uni-stuttgart.de; Tel.: +49-711-685-66350

**Abstract:** Managed retreat has become a recommended adaptation strategy for hazard-prone coastal cities. The study aimed to improve considerations for the contextual factors that influence the success of managed retreat and resettlement projects in Metro Manila. Data were collected through a mixed-method approach consisting of a screening of relevant literature, a qualitative case analysis of resettlement projects, and a workshop series with Philippine stakeholders. It turned out that the resettlement of informal settlers is a central element of urban development. Though in-city resettlement is preferred, the majority of existing and planned projects are developed in off-city locations. The findings present a nuanced view of different retreat approaches. Not all in-city resettlements are successful, and the unpopular off-city projects have a potentially important role for urban and regional development. A strategic planning thread to develop concepts for qualitative off-city settlements that counteract uncontrolled urban sprawl with monofunctional residential areas for urban poor people was deduced. The other thread asks for pathways for inner-city development with innovative, vertical, in-city projects. A final observation was that climate change and the COVID-19 pandemic are worsening the situation in informal settlements, thus strengthening the argument for the planned decentralization of Metro Manila's congested urban areas.

**Keywords:** managed retreat; resettlement; climate change; hazards; informal settlements; urban development; COVID-19; Metro Manila



## 1. Introduction

Metro Manila or the National Capital Region (NCR) is one of the several fast-growing, hazard-prone, mega-agglomerations in Southeast Asia. As the Philippines is listed among the 10 countries most at risk of extreme natural events [1], its densely populated NCR faces a variety of hazards, with typhoons and floods as the most frequent and devastating. Climate change amplifying the existing hazard risks, and socio-economic development with fast urbanization is also exacerbating the situation critically [2–4]. In particular, the people living in informal settlements, which are estimated to account for more than 580,000 informal settler families (ISFs) or roughly three million individuals in the NCR [5,6], are the most exposed because many are living in or along danger areas such as waterways. These informal settlements exist in parallel to the formal system and do not comply with planning regulations, land-use systems, and laws that are meant to build and safeguard resilient settlements and systems [7]. Thus, these settlements only dispose of little or no risk-reducing infrastructure, consist mainly of low-quality housing, and provide limited capacity to cope with stresses [8–10].

One adaptation measure for these informal risk-places is managed retreat. The Intergovernmental Panel on Climate Change (IPCC) named retreat as one of three major

adaptation mechanisms of urban coastal areas exposed to hazards, with protection and accommodation as the other two (IPCC 2014). Retreat is neither a low-regret strategy nor a post-event reaction [11] but the planned, managed, and permanent movement (retreating) of people and/or infrastructure away from hazard-prone areas to reduce hazard exposure and, ultimately, the hazard risk. It is an anticipating instrument of land-use planning or governmental program that seeks to render people and space more governable [12]. In theory, retreat is referred to as an option of last resort that only becomes a justifiable strategy when the other adaptation options are not practicable anymore. The reality looks different, at least in Metro Manila, where retreat is an often-applied strategy, albeit mostly not under the label of retreat but in the form of resettlement projects. Accordingly, retreat must be understood as the general political strategy and resettlement or relocation as its practical components. To prevent misunderstandings, the terms resettlement and relocation are used interchangeably in this study, as this is the case in many political agendas in the Philippines. Both terms are defined in this paper as "physical movement of people to a new place to live other than the previous place" [13].

This study draws attention to planning and policy implications regarding managed retreat by mediating between the conflicting notions associated with both hazards and resettlement. This implies that it certainly requires planning tools and strategies addressing the urban poor, as they are often the hardest hit by disasters and have to live under conditions of everyday risk [14]. However, the potential risk-reducing effects of managed retreat are only one side of the coin. On the other side, it is feared that eviction and the movement of people, which will occur in the wake of retreat, are a trigger rather than a cure for risk [11]. In fact, decades of research on forced migration, displacement, relocation, and resettlement have revealed that—if any—only very rare examples of successful projects exist, while there is a substantial likelihood that the relocated people are impoverished [12,15–19]. In light of this broad knowledge base on potential impoverishment associated with relocation, the authors of this paper investigated the practices and their effects in Metro Manila. This is crucial, particularly in light of the recent policy focus that intends to resettle hundreds of thousands of people who are living in danger areas and the expected aggravation of the caused by the impacts of climate change that will expose additional large numbers of people to danger. Accordingly, the main aim of the paper was to systemize and improve the understanding of managed retreat and associated resettlement practices in Metro Manila. It therefore contributes to the ongoing discussions on lessons learned from decades of research on failed resettlement projects, which is important because large scale resettlement activities under the banner of managed retreat are expected to be unavoidable in the course of climate change [11,12,18–22].

## 2. Research Approach and Methods

This paper was developed to present findings on managed retreat from the research project titled Linking Disaster Risk Governance and Land Use Planning: the Case of Informal Settlers in Hazard Prone Areas in the Philippines (LIRLAP). It followed an iterative research design with a mixed-method approach to meet the main research objectives of systemizing the knowledge of managed retreat in Metro Manila and improving the consideration of the context factors that influence the success of resettlement. The iterative nature of the research was linked to research questions, which guide the subsequent sections of the paper. Sections 3–5 elaborate the conditions in informal risk places, present an overview of existing retreat practices and approaches in Metro Manila, and systemize these approaches in the form of a typology, respectively. The following research questions are discussed:

- Who are the targets for managed retreat in Metro Manila and in which conditions do these people live?
- Which general retreat practices and approaches exist?
- What are the key components of resettlement projects and which characteristics do these components have?

Subsequently, Section 6 conceives lessons learnt regarding the introduced resettlement types and their potential effects on urban development processes. The paper concludes with Section 7 by providing a perspective on progressive ways forward regarding managed retreat in Metro Manila. This section discusses possible future developments and areas of research, blind spots, and central strands of the discussion.

The methodological approach chosen for this iterative research process consisted of a screening of relevant studies on managed retreat, qualitative case analyses of existing resettlement projects, and a workshop series with discussion panels, field visits, and survey and validation activities. By applying these methods, it was possible to look at retreat from various perspectives and to disassemble resettlement practices into their components. These characteristic components allowed us to distinguish different resettlement approaches and, therefore, the formulation of the resettlement typology. The typology was presented to the stakeholder and validated during the second workshop. Admittedly, the research was severely hampered by the COVID-19 pandemic. Planned fieldwork in resettlement sites and physical meetings were not possible in large part due to the travel restrictions and implications of the lockdown.

### 2.1. Screening of Relevant Studies on Managed Retreat in the Philippines

An extensive screening of studies on managed retreat in the Philippines was undertaken as the basis for the research. This screening included the large body of international research on displacement and forced migration. These publications provide insights from decades of research on the negative and positive effects of various forms of resettlement, livelihood outcomes, and the risks of impoverishment, like previous and recent work on development-induced displacement and migration, which has often focused on specific projects such as the construction of large dams all around the world [17,23–27]. Furthermore, the screening included analytical frameworks such as Scudder and Colson's stage model [15,28] and Cernea's impoverishment risks and reconstruction (IRR) model [16]. This work laid the foundation for the theoretical and conceptual thoughts of the rather newer body of literature, which focused on planned resettlement and therefore discussed the concept of retreat [11,12,18–22]. Climate change can be regarded as the central driver for this recent and ongoing discussion. The analyzed literature can be divided into the following general categories:

- Relevant literature on managed retreat, relocation, and resettlement.
- Policy documents, legislations, government agendas, and official strategy papers from the Philippines.
- Project-related documents from different stakeholders including national government authorities, international organizations, and non-government organizations.

### 2.2. In-Depth Case Analyses of Individual Resettlement Projects

The screening of official project-related documents led to an initial list with existing and planned resettlement projects. This initial list was further enlarged by projects that could be detected through a screening of satellite images. These projects identified via satellite imagery were mainly detected based on their building structures, meaning that settlements were marked as potential resettlement sites when they revealed structures characteristic for most of the already known sites. These characteristic structures included orderly street layout, large and monofunctional residential areas, and settlements with clear borders. Examples of this time-consuming detection procedure can be seen in Figure 1.

All detected potential sites needed to be verified through further specific document and internet research based on the known location and administrative affiliation. By this means, a project list of almost one hundred existing resettlement projects within the NCR and the adjacent regions Calabrazon and Central Luzon was developed (Appendix A—Figures A1 and A2).

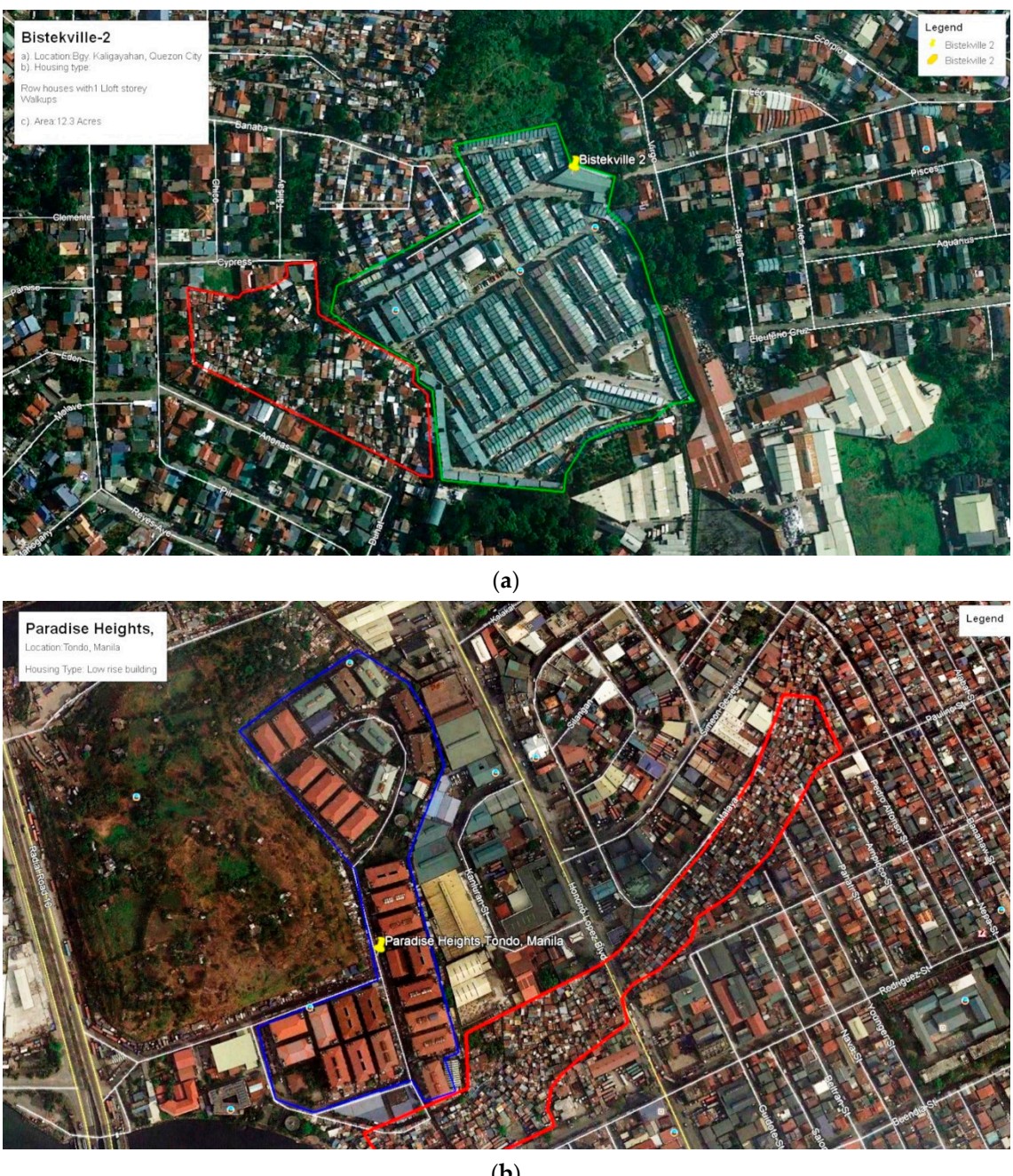

(**a**)

(**b**)

**Figure 1.** Examples of detected in-city resettlement sites with adjacent informal settlements marked in red: (**a**) Bistekville 2 in Quezon City and (**b**) Paradise Heights in Tondo, Manila. Source: Project team design (Megha Kanaginahal) in 2020, Satellite image: Google Earth 2020.

The project list served as the starting point for in-depth analyses of individual projects. In this discourse, several projects were selected to be studied in three steps. The first step consisted of the further exploration of available official documents and information on the project. In the second step, GIS-mapping activities and spatial calculations were conducted. This included the assessment of the location and surroundings, as well as calculations for the size and extent of the area and the population density, the last of which was calculated based on official information on the number of the inhabitants. The last and third step was a qualitative analysis of the project that investigated the involved stakeholders, the built environment (including the housing types, the provided facilities, and the provided services), and the development stages and participation options. Care was taken to ensure

that a variety of project types were screened with this threefold approach to do justice to the diversity of resettlement. The in-depth case analysis was central to understanding resettlement approaches and practices, as well as to extract the essential components for developing reoccurring types. The GIS-mapping activities further gave an initial idea of the spatial distribution of projects and the regional integration into urban and rural systems.

*2.3. Workshop Series with Pre-Survey and Validations*

Significant components of the applied methods were two conducted workshops that were participated in by key stakeholders. In each workshop, around 50–60 stakeholders participated in the four-day activities. The participants were selected to represent relevant institutions; among them were officers and representatives from national government agencies (for example, the Department of Human Settlements and Urban Development (DHSUD), National Housing Authority (NHA), Social Housing Finance Corporation (SHFC), Department of Public Works and Highways (DPWH)), from the Metro Manila Development Authority (MMDA) as a regional agency, from local government unit administrations (LGUs), from international organizations such as the Asian Development Bank (ADB) and UN-HABITAT Philippines, and from non-government organizations and academic institutions. The level of experience ranged from an undersecretary and officer-in-charge to junior researchers of these agencies and institutions.

The first workshop was physically situated in the Philippines and included field visits to selected representative resettlement sites. On the agenda were a stakeholder analysis and the compilation of relevant policies and programs that could be deduced in plenary and small group discussions. The second workshop could only be conducted virtually due to the coronavirus pandemic. It included an online pre-workshop survey and was mainly carried out in the form of presentations, question and answer sessions, and interactive real-time validations. Given the number and level of knowledge of the participants, the pre-workshop survey was developed as a qualitative expert survey. It was sent prior to the workshop and included a variety of questions including seven-point Likert scale ratings, multiple- and single-choice questions, and open questions. A total of 43 responses were received, but only 21 respondents completed the survey and answered all 25 questions. The Mentimeter tool was used during the workshop to incorporate the participants in real-time and to validate the presented findings (for the detailed results—see Supplementary Material). Again, seven-point Likert scale rankings were used to assess the level of agreement or disagreement with the findings, whereas multiple-choice, single-choice, and open questions were raised to further investigate opinions and detailed perspectives.

## 3. Risk Places—Informality in Metro Manila

As mentioned, it is estimated that up to three million people of the well-over 16.5 million inhabitants of Metro Manila live in informal settlements [5,6]. This is significant because the Philippines has witnessed strong economic growth of an average of above 5% in the last few years including job creation, which led the low rate of 6% unemployment in urban areas in 2016 [6,29]. Nevertheless, there has been little success regarding poverty reduction. Poverty decreased by only 1.3% between the years 2012 and 2015, which is a sign that poverty in the Philippines is less a result of joblessness but rather of low earnings [6]. Rutkowski expresses the situation similarly as he identifies the prevalence of bad jobs, which are "low-paid and informal, and thus, not covered by labor regulations" [29]. These precarious conditions persist in rural and urban areas. Nationwide, more than 75% of all workers are employed informally [6,29]. While the Philippines is one of the fastest urbanizing countries in East Asia, the country has not benefitted as much from positive urbanization effects as other countries [30]. In effect, urban areas have attracted a mainly low-skilled and low paid workforce, which has led to an urbanization of poverty. Escaping from this poverty trap is challenging. Singh and Gadgil's study [6] on informality in Metro Manila revealed that the majority of the slum population have lived in Manila since childhood. This implies that scapegoating rural–urban migration as the central trigger for

poverty and growing informality is misleading and underrating failures in urban planning and land policies aimed at generating affordable housing [6].

The lived reality of informality is as ambiguous as the individual settlements are. Informal settlers are not only income-poor, as poverty is rather multidimensional. Slum dwellers experience fundamental deprivation in areas such as sanitation, health services, security of tenure, and education [6]. However, informal settlements are not necessarily places of absolute poverty and deprivation. Manifold formal–informal relationships exist [31,32], and people who are employed formally, who have achieved academic degrees, or who are rather associated as lower-middle-class—for example, governmental workers—are living in informal settlements [6]. Sometimes, living in informal areas with the prevailing unclear or illegal status and lower level of service provision is a pragmatic decision. Proximity to the workplace, for example, might be considered more important or the houses and apartments within the formal city might simply not be affordable due to skyrocketing prices [9,32,33]. It was reported by workshop participants that there is a thin line of informal and formal status, as well as that tenure is a continuum rather than a fixed term. The significance of this becomes apparent when considering that informal settlements are often overseen, ignored, or even accepted due to political decisions [33]. This is largely due to the fact that informal settlers are also voters and local politicians on the barangay and LGU levels who have a short political cycle of only three years, and they tend to make promises and concessions to ISFs. Together with the enormous housing backlog of at least one million missing units [34], these are key factors for the sustained prevalence of informal settlements. Public spending on the housing sector is very limited despite these challenges and their social implications. The government only allocated 0.1% of the gross domestic product for the housing sector between the years 2000 and 2014, which was the lowest percentage in Southeast Asia [5,30].

Informal settlements can be found in various forms throughout all 16 cities and one municipality (Pateros) that form the administrative area of Metro Manila. There are high-density informal settlements and low-density ones with more open space, there are linear informal settlements along waterways and railroads, and there are so-called pocket-settlements that have grown in small vacant spaces [6]. Of major importance to this study, however, were the settlements that are highly exposed to threats of natural hazards. The poor settle in extremely exposed areas of flood plains, on steep slopes, along rivers and the coastline, or even on the waterways [6,9,35]. The LGUs of Metro Manila have altogether identified 104,000 ISFs or around 18% of the total number of ISFs who live in such danger areas—including exposed areas next to railroads or streets. However, the prevalent monsoon rain, typhoons, potential earthquakes, and poor drainage systems suggest that it is not just the detected 104,000 ISFs who are directly exposed to natural threats, because many additional ISFs with low-quality housing are also exposed. This is the case because, for example, the danger zones along waterways are defined by the Philippine water code as the area within the three-meters easement of the waterway. Floods in the wake of typhoons, however, have flooded vast areas beyond the direct riverbanks. Against the background of the anticipated effects of climate change, such as sea level rises and the further intensity and frequency of extreme events, the number of highly exposed ISFs and therefore targets for retreat is most likely significantly higher than the above-mentioned number of ISFs living in danger areas.

## 4. Managed Retreat in Metro Manila—Practices and Approaches

In contrast to the sometimes laissez-faire handling of illegal settlements exists a long tradition of attempts to remove slum areas and to address the housing crises. Approaches include criminalization, forced eviction, relocation, the provision of socialized housing, the upgrading of structures, and the granting of land rights [9]. However, all of them have not been able to tackle the elaborated root cause of an "economic system that depends on cheap labor but cannot provide for adequate housing" [9]. Resettlement as a policy approach appeared mainly from the 1960s and 1970s onwards in a centralized manner, and

in 1975 the National Housing Authority was established as the main agency to implement resettlement and to provide housing for underprivileged, poor, and homeless [32,36]. During that time, massive top–down relocation schemes were the norm and created satellite settlements outside urban areas [32]. A turning point of urban politics was the People's Power Revolution in 1986, which led to a decentralization of governance structures [37]. Relevant laws in this respect are the Local Government Code of 1991 and the Urban Development and Housing Act of 1992 (UDHA). In particular, the UDHA mandates the broad participation of beneficiaries of social housing projects and encourages their involvement through community-based approaches [38]. It also states that on-site developments are preferred and resettlement thus only be executed when such on-site projects are not possible (Section 28 of the UDHA). Eviction and demolition in the course of resettlement activities are to be minimized, and, if necessary, a clear technical guidance is to be provided (Section 28 of the UDHA). In a similar realm does the NHA itself describe the preference for in-city strategies or resettlement locations in close proximity to the previous settlement "to minimize the social, economic, cultural, and political impacts of dislocation" [39]. Though the legislation is supportive for innovative participatory in-city projects and a strong civil society with the formation of urban poor groups, the resettlement provision-scheme of large-scale off-city projects has not changed significantly [32,40,41].

Concerning the concrete reasons behind resettlement activities or the selection of beneficiaries, it can be observed that the general motives to remove slums were initially to address insecure land tenure and to rehabilitate public areas or to support private interests [42]. Further important reasons have been development projects such as highway construction or the metro lines, as well as post-disaster resettlement for calamity affected people. More recently, environmental protection, disaster risk reduction, and climate change adaptation have evolved as new reasons [34,40]. Milestones for this development include the landmark decision of the Supreme Court to clean the Manila Bay. This mandamus decision included the order to actually implement the UDHA from 1992 and therefore clean the danger zones along the three-meter easement of all waterways that drain into the Manila Bay from illegal structures [43]. The second milestones were the devastating super typhoons Ondoy (international name: Ketsana) in Metro Manila in September 2009, Yolanda (international name: Haiyan) in Central Philippines in November 2013, and the just-concluded Ulysses (international name: Vamco) in November 2020. The impacts of these disasters have profoundly changed the perception of hazard risk, the potential amplifying effects of climate change, and the role of illegal settlements [44,45]. The Disaster Risk Reduction and Management Act of 2010 was, for example, a direct answer to typhoon Ondoy [46]. As a consequence of these decisions and experiences, policy regulations have been increasingly targeting the hundreds of thousands of informal settlers living in danger areas [40,42,47] The planned and implemented resettlement of these people is anticipating disaster risk and climate change impacts and can thus be regarded as managed retreat activities, although they are officially not labelled as retreats [48].

A prominent example of a concrete managed retreat policy program is the 50 billion peso housing program launched between 2011 and 2016, which aimed at relocating all of the identified 104,000 ISFs from the major waterways. The people who lived and who are partly still living in these settlements are at high risk of losing property and life in cases of hazards. Additionally, they pollute and often even block the waterways with their constructions and building materials. In case of a flood, these blocked waterways hinder the immediate water discharge and could therefore cause the flooding of adjacent streets and quarters. Apparently, a retreat provides a suitable option to clear areas of ISFs while also tackling pollution problems, introducing floods protection measures by unblocking the waterways, and resettling vulnerable people to less exposed areas.

Regarding the changing context conditions of urban development processes, which are altered by climate change and interlinking trends such as further fast urbanization and socio-economic polarization [11,18,19], it is rather likely that large-scale resettlement programs, such as the 50 billion peso program, become recommended practices—thus

making managed retreat a preferred adaptation strategy. Metro Manila faces an unprecedented scale of people who will potentially be relocated and of resettlement projects that will shape the outskirts of the mega-agglomeration. Broadly speaking, in Metro Manila, it is no longer merely a question of resettling some hundred families who live in shanties where, for example, the new motorway extension is planned. It is rather about resettling a significant share of the almost three million people living in informal settlements. This is tantamount to massive movements of people, potentially shifting the social fabric of the cities.

## 5. Typology of Retreat in Metro Manila

Comparative studies have addressed the diversity of strategies and general approaches to retreat around the world [11,20]. Efforts to organize retreat in the course of these studies have focused either on a superordinate level, such as the study of Hino et al. [11] that developed a framework by focusing on the questions of who benefits from relocation and who initiated the move, or they have focused on the concrete planning approach and thereby investigated the temporal and spatial dimension of retreat [49]. The temporal dimension is related to the temporary or permanent state of the retreat, whereas the spatial dimension examines what will be retreated, including whole settlements, only quarters or parts of settlements, singular buildings, critical or sensible infrastructure, or the reclassification of land-use types which may be later used with less sensible function [49]. These different approaches of retreat might require different policy frameworks, law regulations, and planning mechanisms. Several examples include building restrictions, the development of hazard mitigation strategies, buyouts programs, and tax incentives [20,50]. As elaborated in the previous sections, retreat in the Philippines has mainly been a form of the resettlement of whole settlements, certain quarters or sectors of settlements, and singular buildings that have been either in the way of developing projects or are located in certain restricted or danger areas. This study of the Philippine case was specific, as it dealt with the relocation of informal settlements and illegal structures. Mostly, there are either no or unclear ownership, meaning that the structures are occupying land illegally, although they are often partly accepted and consolidated over the years. Accordingly, the legal bases for the eviction of people and the demolition of buildings is enforceable. Given this, retreat in the form of resettlement is sometimes even perceived as an offer to beneficiaries, as they are provided a new place to live while they are getting loan offers or livelihood assistance.

On the one hand, the analysis revealed that physical attributes such as the location and the housing type characterize resettlement. On the other hand, policies, strategical processes, and finance mechanisms also play a decisive role and allow for the distinguishing between different types. Thus, the present typology for resettlement sites extended the characteristics beyond the physical structure of the built environment and differed from approaches such as applied by Singh and Gadgil [6], who performed slum mapping activities in Metro Manila based on build environment characteristics.

The following four typology components were decided to holistically characterize resettlement approaches in Metro Manila: (1) location, (2) program and finance, (3) strategy and participation, and (4) housing.

### 5.1. Location Component

The location of the resettlement site is the central distinctive feature in the public and scientific discussion for retreat activities in Metro Manila. The discussions particularly focus on the dichotomy between the poles of in-city, perceived as the favorable urban location, and off-city, perceived as a location outside urban areas and away from livelihoods and infrastructure. These two and the third category of near-city were officially defined by the newly developed Department of Human Settlements and Urban Development, under which the NHA operates. In this definition, an in-city resettlement refers to a relocation site within the administrative borders of the same LGU where the ISFs live. A near-city resettlement implies a relocation to a site within the jurisdiction of an LGU that is adjacent

to the LGU where the ISFs have their settlement. Lastly, an off-city resettlement is defined as a site outside and not adjacent to the LGU where the affected ISFs live [51]. These official categories reflect legal and administrative aspects. This explains why the often-disregarded category of near-city remains relevant, as was also supported by the respondents of the pre-workshop survey, who highlighted the relevance of all three categories. Accordingly, near-city and off-city imply that two different LGUs are involved in the process: the sending and the receiving LGUs. This means complicated agreements and compensations, such as an LGU having to finance services like school infrastructure and other material for the mostly poor new residents.

In addition to the administrative aspects, the distance of the previous informal settlement to the new retreat settlement and the level of urban services are the other factors that distinguish between resettlement location types. For the relocatees, these are the important factors, as revealed by the pre-workshop survey. Thus, the three official categories do not sufficiently reflect the importance of distances, because LGUs can be large. The description of near-city, for example, can tell relatively little about the urban quality and the distance between the old and new settlements. On the one hand, it can imply a relocation to a rather peripheral area some 20 or more kilometers away in an adjacent LGU, but on the other hand, it can imply a relocation from downtown Quezon City to downtown Marikina City.

Consequently, we developed a scheme to explain the location categories by considering the legal and administrative aspects, the distance, and the urban characteristics. This nuanced understanding is demonstrated in Figure 2. In addition to the official three categories, two new categories were introduced—in-Barangay and near-site. In-Barangay covers a relocation within the smallest administrative scale and therefore specifies the mostly tight social bonds within a quarter. Near-site implies the relocation in a very short distance, such as to a less exposed area on the other side of a road. Near-site is distinct to on-site activities, which are a form of settlement upgrading. Both additional categories reflect the spectrum of in-city.

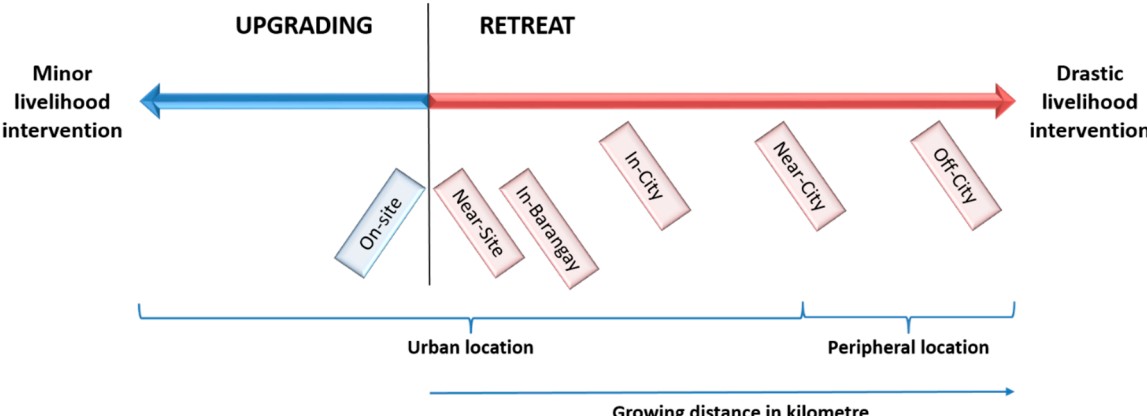

**Figure 2.** Upgrading–retreat continuum depicting the location types and their relation to livelihood intervention, distance, and urban services.

Concerning urban services, a relevant differentiation was made between a location within the agglomeration and in the outskirts of Metro Manila, including the neighboring rural areas. Accordingly, the two overarching characteristics of urban location and peripheral location were developed. Off-city was assigned as peripheral, whereas in-city, with its further distinction in near-site and in-Barangay, was assigned as an urban location. Near-city needs closer attention and can be both peripheral or urban. For the resettlement location categories, the greater the distance to the former settlement is, the more peripheral the new location is. The more peripheral a location is, the fewer urban services and employment opportunities it offers and the more it moves people away from their social networks. This is tantamount to a more drastic intervention into people's livelihoods.

## 5.2. Program and Finance Component

The component of program and finance covers deeply political characteristics of resettlement. This is obvious for political programs, but the financial aspects are also political because they interfere with mechanisms for social housing and cooperatives.

The programs mainly adhere to the NHA as the main agency. The NHA has different programs in its portfolio. These are the resettlement programs for ISFs affected by infrastructure and those living in and along the danger areas. There is further a program for calamity victims and a regional resettlement program. Additionally, there are an upgrading program and a program for vertical development, implying low- and medium-rise buildings with up to 15 floors. Besides the official programs, there is also a distinction criterion on the delivery scheme. The NHA acts either as direct deliverer and developer of settlements; it acts as a member in joint ventures with private landowners, civil society organizations, or LGUs; or it supports incremental housing projects. Joint ventures are mainly realized when the initiative comes not from the national authority but from such organizations or LGUs who have their own shelter targets and resettlement activities.

From the moment when ISFs become beneficiaries of a resettlement program, they are inside an official loan or rent-scheme. Though some ISFs pay rent in informal settlements, being a beneficiary of a resettlement program implies additional financial burdens for most ISFs, as the payments within the government programs are higher, particularly for all in-city locations. Mostly, resettlement loans must be paid in monthly rates for 30 or 40 years. After this time, either the beneficiaries own the land and the house, or they own only the house but not the land in usufruct systems. The challenge for the ISFs, like for most urban poor worldwide, is how to finance housing and loan payments. It is particularly challenging for ISFs who work informally because they do not have access to the traditional financial system because this system targets wage earners in the formal labor market and individual borrowers [52]. Accordingly, the people who work in the informal economy are normally not members of financial services such as the Government Service Insurance System (GSIS), Social Security System (SSS), and the Filipino saving scheme PAG-IBIG Fund (officially known as the Home Development Mutual Fund), and many ISFs do not even have a bank account.

To buffer this weak position of urban poor, various housing loan schemes have been set up. For example, the National Home Mortgage Finance Corporation has created a secondary market for loans that allows poor people to access finance mechanisms. However, the most common way to finance informal settlers who are outside the formal financial system is the Community Mortgage Program (CMP) as the flagship program for socialized housing. The central idea behind this program is that mortgage loans are not provided to single persons or families but for communities of poor people. Such community mortgages make it possible for poor to access financial support and thereby acquire security of land tenure, either on the site they are living (upgrading) or the site they are intended to be relocated. The program also encourages landowners to sell their land to urban poor communities by providing incentives such as offering exemptions from payment of capital gains tax. The approach is highly locally-driven, as the community has to organize itself, investigate possible land, and be involved in the design of projects and long-term maintenance.

## 5.3. Process and Participation Component

Resettlement is not a short-term process. As visible in Figure 3, it can be subdivided into three phases of project development, lasting, on average, more than one year and a fourth long-term phase for monitoring and estate management [40]. Phase I is the pre-relocation or social preparation phase. It consists of the identification of beneficiaries and resettlement sites, as well as the mobilization of resources [40]. It is a phase of intensive planning and negotiations, including calls for tender and bidding processes, and can entail very bureaucratic processes whereby the principle applies that more bureaucratic hurdles of legal and administrative issue exist for near-city and off-city projects because at least two

local authorities, the sending and receiving LGUs, are involved. Additionally, the more intense the participation is, the longer phase I is. For example, the building of associations in community-driven approaches is a complicated process. Further complicated is the identification of sites in these community-driven projects, particularly because they mainly focus on the rare near-site, in-Barangay, and in-city sites.

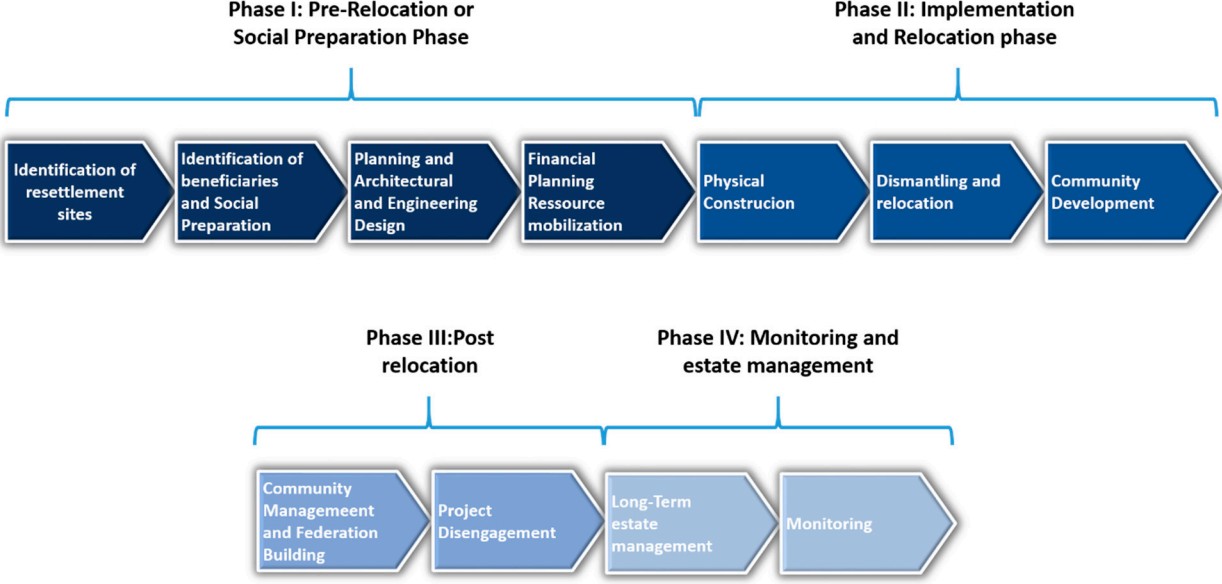

**Figure 3.** Four-phase model for resettlement in the Philippines.

Phase II is considered the physical phase. It starts when the involved agencies and developers have signed contract agreements. It includes the physical construction of the sites and the dismantling of the old settlements. Subsequently, the actual movement of people can take place. Ballesteros and Egana [40], who reviewed the efficiency and effectiveness of NHA resettlement programs, emphasized the required duration of approximately one month to relocate 1000 beneficiaries. Phase III is the post-relocation phase and starts from the moment when the relocation operation is completed. It consists of the important process of mentally arriving at the new settlement and includes the integration of different groups into a federation. Phase III marks the end of the project development and, therefore, ideally the disengagement of some actors. The transition from phase III into the monitoring and estate management of phase IV is fluid. In fact, phase IV has no defined beginning and end. It is often not even calculated and considered as part of the resettlement process. However, the phase is crucial because it entails important services such as livelihood support programs and the management of the facilities. Furthermore, it is necessary to state that the disengagement of the NHA or other leading agencies after phase III has often not been possible. The NHA has to collect the monthly amortization, which has to be paid by the beneficiaries of various housing loans, while the receiving LGUs often need technical and financial support for services in the new communities [40].

When asked which phase is most important for the success of resettlement projects, the experts who participated in the pre-workshop survey signified that they consider all phases as equally important. In reality, however, the focus is on the planning, implementation, and delivering processes of projects, whereby phase I is considered the most critical [34]. It is certainly true that the prospects for successful projects become more promising the more thoroughly and detailed the planning and implementation of phases I, II, and III are. However, following the findings of the work of Scudder and Colson [15,28], the focus on the first phases might undermine the important long-term processes with potential development and incorporation in the new settlement, which are often or only achieved after several years.

All resettlement projects officially entail participation activities. Participation mainly takes place in phase I with the identification of the beneficiaries and the social preparation. However, the degree of participation varies substantially. Referring to Arnstein's [53] classical work on participation, the range is between informing and consultation on the one hand and partnership and delegated power on the other hand. The first is a case for agency-driven approaches. The identified beneficiaries are approached by LGU and barangay officials and informed about their coming eviction. There are regulations regarding how the process of eviction has to be conducted and how the affected people have to be targeted. Accordingly, the ISFs are offered options, as expressed by NHA officials during the workshop series. These include the distribution of resettlement project flyers or the organization of community field trips [40]. A fundamentally different level of participation in the sense of distributing power is possible in cases where ISFs unite as associations and work in cooperation with non-government organizations (NGOs), LGUs, and government agencies on people's plans or with the CMP. In such cases, the ISFs are involved or even leading processes along with all resettlement phases. This involvement can be more or less successful and entail more or less intense support from other actors, as described by Galuszka [32]. Challenges occur due to conflicting political interests and cumbersome regulations and bureaucracy. Moreover, conflicts and challenges within the associations reveal that collective approaches are not without challenges.

### 5.4. Housing Component

The built environments of cities and regions are always heterogeneous and influenced by various architectural styles and shifting planning paradigms over time. As resettlement projects are parts of cities and regions, they are also not purely uniform and homogeneous entities. Moreover, they can be large, particularly in off-city locations where they can consist of more than 4000 units that accommodate around 20,000 inhabitants. These large settlements can be composed of a variety of housing types and styles. Furthermore, it is often the case that over time, incremental changes and individual extensions are undertaken by the owners, therefore changing the appearance of the settlements and houses.

However, while appreciating heterogeneity within and between resettlement sites, a typology intends to detect typical structures and generalize recurring housing types. This is possible because resettlement sites are mostly monofunctional, designed to provide shelter for large quantities of poor people. Furthermore, for socialized housing projects, site development has to follow standards such as the Batas Pambansa BLG. 220, which is a law developed to "establish and promulgate different levels of standards and technical requirements for the development of economic and socialized housing projects, and economic and socialized housing units in urban and rural areas" [54]. The characteristic housing types of resettlement sites can be seen in Table 1.

**Table 1.** Resettlement housing types.

| Single Detached | Duplex/Single Attached | Rowhouses | Walkups | Multi-Story Low-Rise Buildings |
|---|---|---|---|---|
|  |  |  |  |  |
| (**a**) | (**b**) | (**c**) | (**d**) | (**e**) |

Sources: (**a**) Single detached and (**b**) and duplex/single attached [55]; (**c**) rowhouses: Google Street View; (**d**) walkups: Lauer 2020; and (**e**) multi-story low rise building: APOAMF accessed via [56].

In the research process, these housing types were assigned to the location types via satellite analysis and through the survey and validation activities within the workshops.

The results revealed that the peripheral resettlement sites showed less diverse housing types than the urban ones, whereas the off-city sites seldom consist of higher structures than one-story buildings (partly loftable); the in-city sites including the near-site and in-Barangay categories showed a variety of types and heights. Accordingly, the peripheral off-city projects could be detected rather easily via satellite imagery, as demonstrated in Figure A3 in the appendix. They were found to mainly consist of rowhouses and, depending on the size of the settlement, single detached and sometimes duplex buildings. In-city, near-site, and in-Barangay settlements mainly consist of multi-story, low-rise buildings, particularly in the wake of walkups and the new vertical housing programs of the NHA.

*5.5. Resettlement Profiles*

A summary of the components is illustrated by the typology profiles of Figure 4, which depicts the characteristics of the location component as either urban or peripheral. The program and finance mechanisms are classified as either targeting the individual in a top–down manner or a group or association by applying a collective and bottom–up perspective. The process and participation features can be based on a direct delivery scheme with completed housing while merely consulting the targeted ISFs or they might allow for incremental projects with intense participation processes in partnership. It should be emphasized that the components of program and finance and process and participation are correlating, with bottom–up programs depending on collective funding and resulting in incremental projects with intense participation. The physical characteristics of the housing component can either be horizontal with one- or two-story structures on individual lots or rather vertical with up to five-stories and condominium buildings.

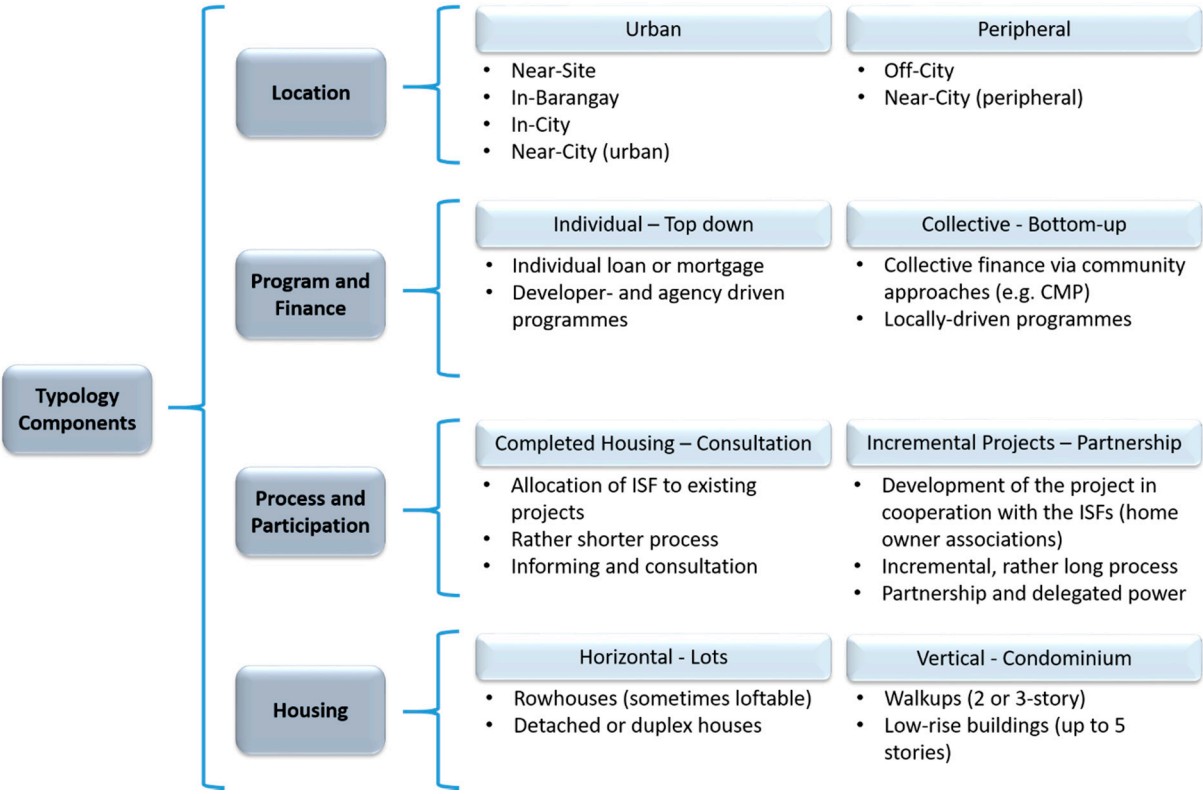

**Figure 4.** Resettlement profiles based on the typology components. ISF: informal settler family.

## 6. Lessons Learnt—Guideposts in Managing Retreat

What are the lessons learnt from the systemizing process and the developed resettlement typology? The following four key assumptions and insights were deduced during the research process.

The first lesson learnt is laying the ground. This emphasizes that the lower the disturbance of the livelihoods of ISFs is in the course of resettlement activities, the better is the outcome of these activities. This insight was derived from the literature on the risks of impoverishment due to relocation [16,19,21,26,27] and from results of the pre-workshop survey. When the experts were asked about the preference of ISFs regarding the resettlement site location, the main answers were living close to the previous income opportunity (occupation) and living close to any kind of income or various income opportunities. It was emphasized in further discussion that ISFs tend to significantly focus on the actual distance of their potential new settlement to their income sources and social networks. This is reflected in the upgrading–retreat continuum in Figure 2. It was argued that any upgrading measures of the informal settlements should be preferred before discussing and planning resettlement options. If resettlement cannot be avoided, a move to available sites close by—to near-site or in-Barangay sites—should be preferred over peripheral near-city or off-city projects.

The second lesson learnt follows this idea of preferred in-city resettlement but is skeptical about the feasibility. We contend that the number of people who live in informal settlements and who are and will be targets for resettlement is very high, so it will not be feasible to resettle those people within the core urban areas. The main reason for this is the scarcity of available land, which, in turn, has its reasons in socio-economic and environmental factors. The socio-economic factors include the demographics situation, with a growing population, a high rate of urbanization with a further influx of people, skyrocketing land prices, and unclear land ownership. Meanwhile, the environmental factors are represented by physical conditions that limit the availability of space, including the vast coastline, existing swamp areas, various waterways, the huge flood-prone Laguna de Bay, hilly areas in the west, and the presence of active fault lines. Climate change is altering most of these environmental context conditions, thus amplifying land scarcity.

The limited land for socialized housing in the central areas has severe effects on urban development and has led to contested political fields and real estate markets. It is not without reason that the majority of already existing and planned resettlement projects in Metro Manila were and are developed in a large-scale manner in peripheral locations [30,32,40,41]. As the housing backlog remains immense, with an estimated one million and more missing units in Metro Manila [34], the high land prices are a challenge for all urban groups, not only for the extremely poor and those who live in informal settlements. The pressure on empty spots and potential building plots remains high and is aggravated by the further influx and even reflux of ISFs [34]. The majority of the experts who answered the pre-workshop survey estimated that the number of ISFs who occupy the danger areas has not significantly decreased or might even be higher than the 104,000 ISFs who had been identified in 2011. This is mainly due to reflux of people when areas have been cleared of ISFs. One expert saw the responsibility by the respective Barangays who often fail to enforce no-build zones in cleared areas or, even worse, tolerate the influx of new or the same ISFs in these areas. This opinion refers to the well-known practice of professional squatting applied by squatting syndicates.

The third lesson learnt was built on the previous ones but also qualifies the first lesson learnt. We argue that not all in-city projects are successful and that the apparently unpopular off-city projects are equally important for urban development. Certainly, the location of a resettlement site is one of the most critical factors with major influence on the resilience-building of the resettled people. However, the physical location is not the only decisive characteristic to influence the success of managed retreat, as can be seen in the expert's rating of Figure 5. Other factors and resettlement components, such as the strategic approach of the involved stakeholders, the financing mechanisms in place, the housing

typology, and the participation possibilities also influence the outcome of projects. Thus, the very general assumption that in-city projects automatically generate better livelihood outcomes while ISFs in off-city or peripheral near-city projects are less resilient and face impoverishment falls short and neglects the complexity of urban development processes.

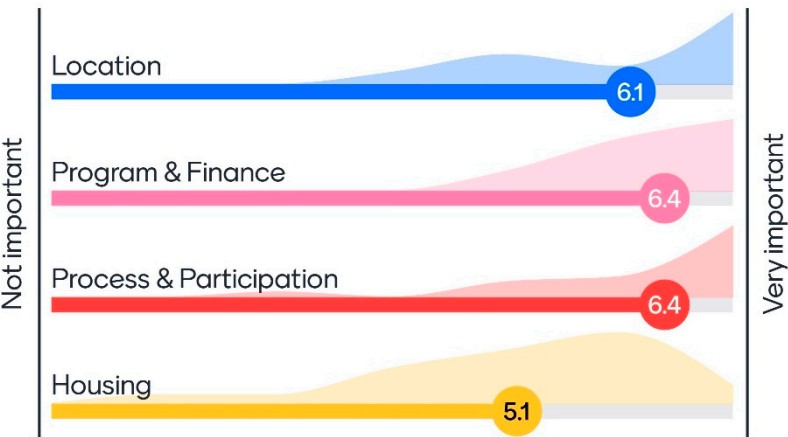

**Figure 5.** Real-time rating during the validation workshop. Results on the importance of the different components for the success of resettlement projects. We used seven-point Likert scales with the Mentimeter tool; *n* = 22.

One reason for the failure of this general assumption is that many ISFs struggle to afford the payments for the more expensive in-city settlements. The monthly arising expenses that have to be paid by the beneficiaries of NHA projects start with 200 pesos in off-city projects (reaching up to almost 1400 pesos after 30 years, depending on the interest rate), whereas in-city projects start with 600 pesos (reaching up to around 5000 pesos, depending on the interest rate and the floor area). A further reason is the housing typology. The multi-story condominium housing in in-city projects can be both an opportunity and a threat. They can be an opportunity because they offer more ISFs lives within urban areas. They can be a threat because they can have manifold unwanted side effects, particularly if they are implemented in a top–down and mass construction manner. In other words, multi-story mass housing schemes in in-city locations might not be a significantly better alternative to completed mass housing off-city schemes. Galuszka argued that the formal morphology of top–down multi-story housing schemes clashes with "the spatial knowledge of informal settlers relocated to those contexts" in these projects [32]. Accordingly, ISFs might prefer horizontal housing types such as row-houses with a little lot over vertical condominium types. Furthermore, the development of ownership during the resettlement process is critical for projects. Galuszka [32] compared the participation process of two projects with people's plan participation. He concluded that the near-city multi-story housing project in San Jose Del Monte just outside Metro Manila was developed with intensive and well-implemented participation. The compared in-city project struggled with similar processes, because due to higher land value and political interest, the project was heavily conflicted and participation processes were complicated. Thus, the near-city projects allowed for more in-depth participation and incremental changes for the housing units, and they were able to build a stronger community sense. This comparison revealed that well-implemented near- and off-city projects are, in principle, capable of fostering resilience-building and, in the long run, providing sufficient livelihood opportunities. Such projects have the potential to develop the outskirts of Metro Manila in a controlled manner and guide the decentralization of the congested urban areas. This was expressed by 16 out of 27 experts from the pre-workshop survey who perceived resettlement as an opportunity to develop the outskirts of Metro Manila, whereas only four perceived it as a risky endeavor that marginalizes the poor at the fringes. Seven experts were undecided and mentioned that it can be a suitable strategy but bears some risk.

These three lessons learnt were recently supplemented by an additional fourth lesson. While investigating managed retreat practices in Metro Manila, the COVID-19 pandemic interfered with the research. The Philippines is the country that has been hardest hit by the coronavirus pandemic in Southeast Asia in terms of confirmed cases and the duration of the lockdown (in October 2020). The socio-economic impacts have been serious, particularly for those already facing poverty and the effects of climate change. Though all the health and socio-economic impacts are not yet foreseeable, the coronavirus pandemic should not be neglected because it directly affects people's livelihoods and the focus of policy programs. Accordingly, the pandemic and its effects were intensively discussed during the second workshop, as can be seen in the expert's rating of Figure 6, and addressed within the pre-workshop survey. Based on this, we argue that the coronavirus pandemic is worsening the situation and the public view of informal settlements, thus strengthening the rationale of resettling poor people and decentralizing the urban areas of Metro Manila.

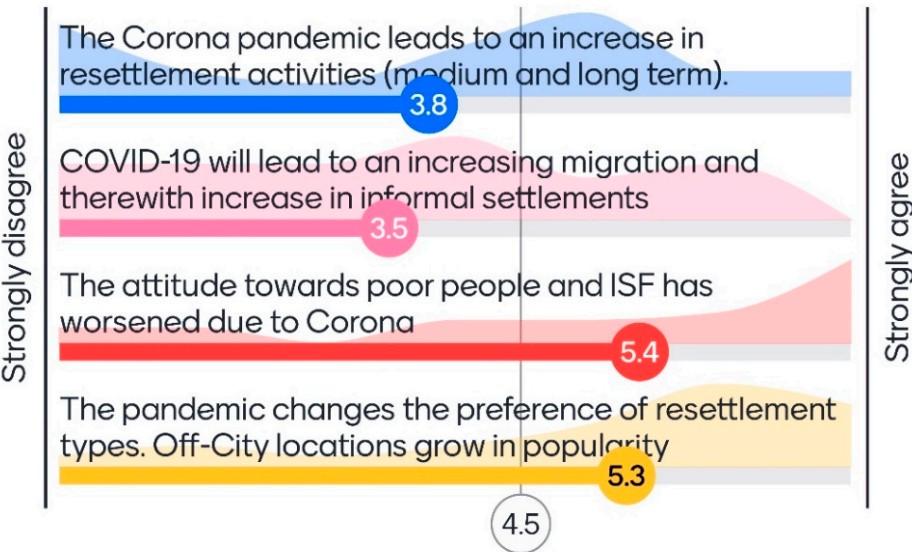

**Figure 6.** Real-time rating results on statements regarding COVID-19. We used seven-point Likert scales with the Mentimeter tool; *n* = 25.

The performed real-time validation tasks of the second stakeholder workshop could have revealed the experts' opinions on the question regarding which poor population groups have suffered most from the coronavirus and lockdown impacts. The experts identified the people living in informal settlements as those who were hardest hit, followed by people in in-city resettlement sites. The people in peripheral near-city or off-city locations were ranked as least severely hit. According to the experts, the lockdown was not as directly enforced and felt in off-city locations, where people could move more freely. However, whereas the direct effects were estimated to be less dramatic in off-city projects, the experts also highlighted the danger of secondary effects, including the interruption of important urban–rural linkages and the limited accessibility of critical services. This might include access to food and goods, as well as health facilities.

## 7. Ways Forward

This paper concludes by providing selected threads for progressive ways forward regarding managed retreat in Metro Manila. The first is that future retreat programs must focus on building resilience. This appears evident against the background of decades of research on possible impoverishment due to relocation, as well as negative examples of failed projects in the Philippines. Following this line of thought, managed retreat must not only reduce the exposure of relocated people from sources of harm but also must be

developmental—this means that, as a minimum, the relocated people should not be worse off regarding their livelihoods [21].

Currently, a functioning indicator-based evaluation and monitoring framework that could assess the specific impacts of different resettlement approaches is lacking. The developed typology could serve as a basis for such a framework. The different resettlement types could be assessed regarding their effects on different resilience dimensions, e.g., the livelihood outcomes. This would significantly improve the development of targeted resettlement projects if it was possible to determine which components of resettlement are responsible for certain effects on resilience-building. Do the missing livelihood opportunities in peripheral locations indeed comprise a crucial factor for making life in off-city projects cumbersome? To what extent are other characteristics like housing type and the quality of the new homes important? If the livelihood outcomes comprise the central factor, can peripheral projects exist with successful strategies that help to create new livelihood sources? Based on an indicator-based assessment, best practice projects might be deduced for every resettlement type and their components.

A second thread concerns matters of integrated urban planning and design. Relocation and, particularly, the off-city projects are major driving forces of urban development in the NCR and adjacent regions. By considering that some of these peripheral projects have been developed for up to 5000 ISFs, which account for roughly 20,000–25,000 people, the designation as a project is somehow misleading. Factually, the implementing agencies are building new communities from scratch. These newly developed settlements must not lead to further uncontrolled urban sprawl with monofunctional residential areas for urban poor people. Such a suburbia of the poor, often developed by a few large contractors, needs to be avoided. Instead, it requires models for qualitative new settlements or new towns that guide matters of decentralization. This could support to reverse rural–urban migration and eventually decongest urban centers—a necessity, as has been recently revealed by the infections of COVID-19, which widely spread, especially within the informal settlement communities. Urban and regional plans are key to detect corridors for further development and regional hotspots where best practice off-city resettlement projects could be located. These projects need to consist of different housing types, densities, and types of tenure that can promote a multiplicity of housing solutions [55].

Simultaneously, it is important to propose new pathways for inner-city development in the form of in-city, in-Barangay, and near-site projects. The further densification of already dense urban areas asks for sensitive and innovative approaches. Vertical projects that meet the needs of dealing with the scarcity of affordable land are wanted, as in an upscaling of innovative financing mechanisms. Both of these require new designs that suit the living conditions of hybrid informal–formal practices and that facilitate mixed developments that can counteract gentrification and segregation processes that displace poor populations from core urban areas [32].

The last thread is the need for mainstreaming, actual implementation, and upscaling. The investigation of policy documents, strategies, and official agendas, as well as the performed case analysis and workshops, revealed a partly well-developed set of regulations, rules, and legislative instructions. The Philippines has developed various development strategies and masterplans that address informality, housing needs, climate change, and resettlement, as well as proposing strategies for integrated action [5,46,57–59]. The relevant agencies have developed policies, and responsible institutions have set regulations, resettlement guidelines, and mandatory building codes for socialized housing [54,60]. As the Philippines is a highly decentralized country, it delegates manifold competences, particularly planning related ones, to the LGUs, who develop their own strategies and programs and who are obliged to establish binding comprehensive land use plans (CLUPs). Furthermore, there has been an evolution of a strong civil society, organizations of the poor, and NGOs who involve in retreat processes [32,61]. Though their influence needs to be extended, they have co-created best-practice community-driven approaches with participation, such as the people's plan possibilities.

By regarding this governance structure with the developed strategies and policies in place, an insight of this study is that challenges regarding informality and retreat exist, to a large extent, in terms of missing implementation and thus in the sphere of everyday politics, of mainstreaming and governance struggles, and interference with the housing market. There are not nearly enough existing best practice examples to address the housing needs, so consistent programs for upscaling are required [9,32].

**Supplementary Materials:** The following are available online at https://www.mdpi.com/2071-1050/13/2/829/s1, Results of the real-time validations of the second workshop.

**Author Contributions:** Conceptualization, H.L. and M.D.R.; methodology and software H.L.; validation, H.L. and M.D.R.; formal analysis, H.L.; writing—original draft preparation, H.L.; writing—review and editing, H.L., M.D.R., and J.B.; visualization, H.L.; project administration, H.L., M.D.R., and J.B.; funding acquisition, M.D.R. and J.B. All authors have read and agreed to the published version of the manuscript.

**Funding:** This paper presents findings from the research project titled, Linking Disaster Risk Governance and Land Use Planning: the Case of Informal Settlers in Hazard Prone Areas in the Philippines (LIRLAP). This project is funded by the German Federal Ministry for Education and Research (BMBF), grant number FKZ 01LE1906C. The project runs until at least 2024 and consists of the four interconnected work packages: Risk Trends, Resilient Upgrading, Resilient Retreat and Capacity Building.

**Institutional Review Board Statement:** Not applicable.

**Informed Consent Statement:** Informed consent was obtained from all subjects involved in the study.

**Acknowledgments:** The authors are grateful to the stakeholders from the Philippines' national government agencies (NGAs), non-government organizations (NGOs), local government units (LGUs), private sector and the academe for their active participation and support in the interviews, surveys, and attendance to the workshops/fora on the Working Package (WP) 3 on Resilient Retreat of the LIRLAP Project.

**Conflicts of Interest:** The authors declare no conflict of interest. The funders had no role in the design of the study; in the collection, analyses, or interpretation of data; in the writing of the manuscript, or in the decision to publish the results.

**Appendix A**

| Site | No. Units (as currently known) | Dominant housing type (further to be validated) | Area_Google Earth in hectare | Density per hectare of G_Earth area | Location |
|---|---|---|---|---|---|
| **National Capital Region** | | | | | |
| Bistekville 1 (Bistekville might be On-Site) | 334 | Walkups | 1.46 | 228.8 | Bgy. Payatas, Quezon City |
| Bistekville 2 | 1078 | Row houses with loft storey | 5.17 | 208.5 | Bgy. Kaligayahan, Quezon City |
| Bistekville 3 | | Row houses with loft storey | 0.18 | 0.0 | Escopa, Quezon City |
| Bistekville 4 | 266 | walkups | 0.96 | 277.1 | Bgy. Culiat, Quezon City |
| Bistekville 9 | 192 | Walkups | 0.43 | 446.5 | Brgy. Gulod, Novaliches, Quezon City |
| Camarin Residences 1, 2 and 3 | 3240 | Low-rise Buildings | 9.61 | 337.1 | Camarin (PCSO), Caloocan City |
| Cardinal Sin Village | 728 | Low-rise Buildings & Walkups | 1.64 | 443.9 | Sta. Ana, Manila |
| Disiplina Village | 594 | Low-rise Buildings | 1.57 | 378.3 | Brgy. Bignay, Valenzuela City |
| Ernestville | 212 | walkups | 0.49 | 432.7 | Brgy. Gulod, Novaliches, Quezon City |
| Manggahan (MMDA Depot) | 120 | Low-rise Buildings | 1.83 | 65.6 | |
| Navotaas 1 Housing Project | 680 | Mixed typology | 7.05 | 96.5 | Tanza, Navotas City |
| Northville 1 | 1299 | Single detached, 1storey loft | 8.4 | 154.6 | Bignay, Valenzuela City |
| Northville 2b | 2184 | Row houses single floor | 20.7 | 105.5 | Caloocan City |
| Paradise Heights | 970 | Low-rise Buildings | 5.43 | 178.6 | Tondo, Manila |
| Pascualer Ville | 994 | Walkups | 3.27 | 304.0 | Brgy. San Bartolome, Novaliches, Quezon City |
| San Juan City Housing Project ph. 1 | 348 | Walkups | 0.55 | 632.7 | Pinaglaban St., San Juan City |
| Southville 3 | 6496 | Single detached single floor | 48 | 135.3 | Muntinlupa City |
| Tala ph. 3 | 420 | Low-rise Buildings | 73.4 | 5.7 | Tala, Caloocan City |
| TBC II | 92 | Row houses single floor | 6.12 | 15.0 | Quezon City |
| **Region III (Central Luzon)** | | | | | |
| Balagtas heights | 1000 | Row houses with 1 storey | 8.13 | 123.0 | Bgy. Santol, Balagtas, Bulacan |
| Balagtas heights expansion | 84 | Row houses single floor | 12.3 | 6.8 | Bgy. Santol, Balagtas, Bulacan |
| Bocaue Hills AFP-PNP Housing | 500 | Row houses single floor | 23.4 | 21.4 | Bocaue, Bulacan |
| Northville 5-A | 1943 | Single detached, 1storey loft | 13.3 | 146.1 | Sta. Maria, Bulacan |
| Northville 6 | 1206 | Single detached, 1storey loft | 17.6 | 68.5 | Balagtas, Bulacan |
| Norzagaray Heights Resettlement | 500 | Row houses with 1 storey | 23.9 | 20.9 | Norzagaray, Bulacan |
| Padre Pio | 247 | Row houses with 1 storey | 22.4 | 11.0 | Bgy. Cacarong Bata, Pandi, Bulacan |
| Pandi Residences 3 | 1000 | Row houses with 1 storey | 22.5 | 44.4 | Bgy. Mapulang Lupa, Pandi, Bulacan |
| Pandi Village 2 | 1000 | Row houses with 1 storey | 13.8 | 72.5 | Bgy. Silling Bata, Pandi, Bulacan |
| Pleasant Hills MRB | 1788 | Walkups | 3.47 | 515.3 | San Jose del Monte, Bulacan |
| San Jose del Monte HTS | 5006 | Single detached, 1storey loft | 53.7 | 93.2 | San Jose del Monte, Bulacan |
| Sapang Palay | | Row houses with 1 storey | 89.8 | 0.0 | San Jose del Monte, Bulacan |
| St. Joseph Ville | 78 | Row houses with 1 storey | 2.89 | 27.0 | Bgy. Kaypian, CSJDM, Bulacan |
| St. Martha Homes | 3790 | Row houses with 1 storey | 51.6 | 73.4 | Bocaue, Bulacan |
| Towerville ph. 6 | 2060 | Single detached, 1storey loft | 20.5 | 100.5 | San Jose del Monte, Bulacan |
| Towerville Ph. 7 | 2000 | Single detached, 1storey loft | 12.2 | 163.9 | San Jose del Monte, Bulacan |
| Villa Elise | 738 | Row houses with 1 storey | 8.31 | 88.8 | Pandi, Bulacan |
| **Region IV-A (Calabarzon)** | | | | | |
| Balete Relocation Site | 849 | Single detached, 1storey loft | 6.64 | 127.9 | Batangas City, Batangas |
| Bayanijuan sa Southville 7 (former Bayanijuan sa Calauan) | 5149 | Single detached, 1storey loft | 24 | 214.5 | Calauan, Laguna |
| Binangonan Resettlement Site | | Single detached, 1storey loft | 1.16 | 0.0 | Binangonan, Rizal |
| Cardona Housing Project | 473 | Row houses with 1 storey | 4.68 | 101.1 | Bgy. Calahan, Cardona, Rizal |
| Consortium | 10000 | Single detached, 1storey loft | 35.8 | 279.3 | Trece Martires City, Cavite |
| Don Jose Homes | 1000 | Row houses single floor | 6.66 | 150.2 | Calamba, Laguna |
| Eastshine Residences | 2000 | Row houses with 1 storey | 17 | 117.6 | Tanay, Rizal |
| Golden Horizon | 2500 | Row houses single floor | 71 | 35.2 | Trece Martires City, Cavite |
| Kasiglahan Village 1 | 3,074 | Row houses single floor | 19.6 | 156.8 | Rodriguez, Rizal |
| Kasiglahan Village 3 | 1000 | Row houses single floor | 28.5 | 35.1 | Trece Martires, Cavite |
| Kasiglahan Village 4 | 1076 | Row houses single floor | 25.4 | 42.4 | General Trias City, Cavite |
| Kasiglahan Village 5 | 1054 | Row houses single floor | 65.8 | 16.0 | General Trias City, Cavite |
| Katuparan Ville Housing Project | 3856 | Single detached, 1storey loft | 69.7 | 55.3 | Tanza, Navotas City |
| Naic Resettlement Site 1 | 1180 | Row houses single floor | 49.2 | 24.0 | Barangay Timalan, Naic, Cavite |
| South Morning View Subdivision | | Row houses with 1 storey | 5.98 | 0.0 | Naic Municipality, Cavite Province |
| Resettlement site in Barangay Santiag | 984 | Single detached, 1storey loft | 37.5 | 26.2 | General Trias City, Cavite |
| Southville 2 ph. 3 | 3999 | Row houses with 1 storey | 13 | 307.6 | Trece Martires, Cavite |
| Southville 3A Ext. | 567 | Single detached, 1storey loft | 13.2 | 43.0 | San Pedro, Laguna |
| Southville 5 | 1822 | Single detached, 1storey loft | 10.2 | 178.6 | Binan, Laguna or Timbao, Biñan? |
| Southville 8b ph. 5 | 605 | Row houses single floor | 65.6 | 9.2 | Bgy. San Isidoro, Rodriguez |
| Southville 9 ph. 3 | 2000 | Row houses with 1 storey | 41.2 | 48.5 | Bgy. Pinugay Baras, Rizal |
| Summer Homes | 1500 | Row houses with 1 storey | 30 | 50.0 | Trece Martires, Cavite |
| Tropical Village | 3246 | Single detached, 1storey loft | 20.5 | 158.3 | Colmenar, Gen.Trias, Cavite |
| Ciudad de Strike 2 | 1440 | Row houses with 1 storey | 5.97 | 241.2 | Bacoor |

**Figure A1.** List of existing and planned resettlement project sites.

| Currently Unknown Sites | | | | | |
|---|---|---|---|---|---|
| **Dominant housing type: Row-Houses** | | | **Dominant housing type: Single detached houses** | | |
| **Name** | **Number of lofts** | **Area(Ha)** | **Name** | **Number of lofts** | **Area(Ha)** |
| Site 1 | Row houses single floor | 12.18 | Site 3 | Single detached single floor | 29.6 |
| Site 2 | Row houses with loft storey | 26.225 | Site 2 | Single detached single floor | 33.6 |
| Site 3 | Row houses with loft storey | 28.02 | Site 3 | Single detached, 1storey loft | 14.6 |
| Site 4 | Row houses single floor | 24.24 | Site 4 | Single detached single floor | 36.4 |
| Site 5 | Row houses single floor | 15.35 | Site 5 | Single detached, 1storey loft | 19.12 |
| Site 6 | Row houses single floor | 8.67 | Site 6 | Single detached single floor | 12.18 |
| Site 7 | Row houses single floor | 20.7 | Site 7 | Single detached single floor | 18.6 |
| Site 8 | Row houses with loft storey | 9.85 | Site 8 | Single detached single floor | 45.95 |
| Site 9 | Row houses single floor | 11.54 | Site 9 | Single detached single floor | 7.1 |
| Site 10 | Row houses single floor | 4.9 | Site 10 | Single detached single floor | 66.9 |
| Site 11 | Row houses with loft storey | 22.7 | Site 11 | Single detached single floor | 57 |
| Site 12 | Row houses with loft storey | 5.04 | Site 12 | Single attached single floor | 23.5 |
| Site 13 | Row houses single floor | 8.38 | Site 13 | Single detached, 1storey loft | 30.2 |
| Site 14 | Row houses single floor | 17.63 | Site 14 | Single detached single floor | 22.43 |
| Site 15 | Row houses with loft storey | 14.55 | Site 15 | Single detached, 1storey loft | 66.79 |
| Site 16 | Row houses single floor | 10.07 | Site 16 | Single detached, 1storey loft | 9.18 |
| Site 17 | Row houses single floor | 6.58 | Site 17 | Single detached, 1storey loft | 12.9 |
| Site 18 | Row houses with loft storey | 4.77 | Site 18 | Single detached single floor | 49.2 |
| Site 19 | Row houses with loft storey | 31.33 | Site 19 | Single detached, 1storey loft | 9.45 |
| Site 20 | Row houses single floor | 4.8 | | | |
| Site 21 | Row houses with loft storey | 7.1 | | | |
| Site 22 | Row houses with loft storey | 4.43 | | | |
| Site 23 | Row houses with loft storey | 18.56 | | | |
| Site 24 | Row houses single floor | 2.96 | | | |
| Site 25 | Row houses with loft storey | 34.33 | | | |
| Site 26 | Row houses single floor | 5.66 | | | |
| Site 27 | Row houses single floor | 4.92 | | | |
| Site 28 | Row houses with loft storey | 12.08 | | | |
| Site 29 | Row houses single floor | 13.39 | | | |
| Site 30 | Row houses single floor | 12.35 | | | |
| Site 31 | Row houses with loft storey | 26.09 | | | |
| Consortium | Row houses single floor | 16.25 | | | |

**Figure A2.** List of sites detected by satellite imagery—still to be verified.

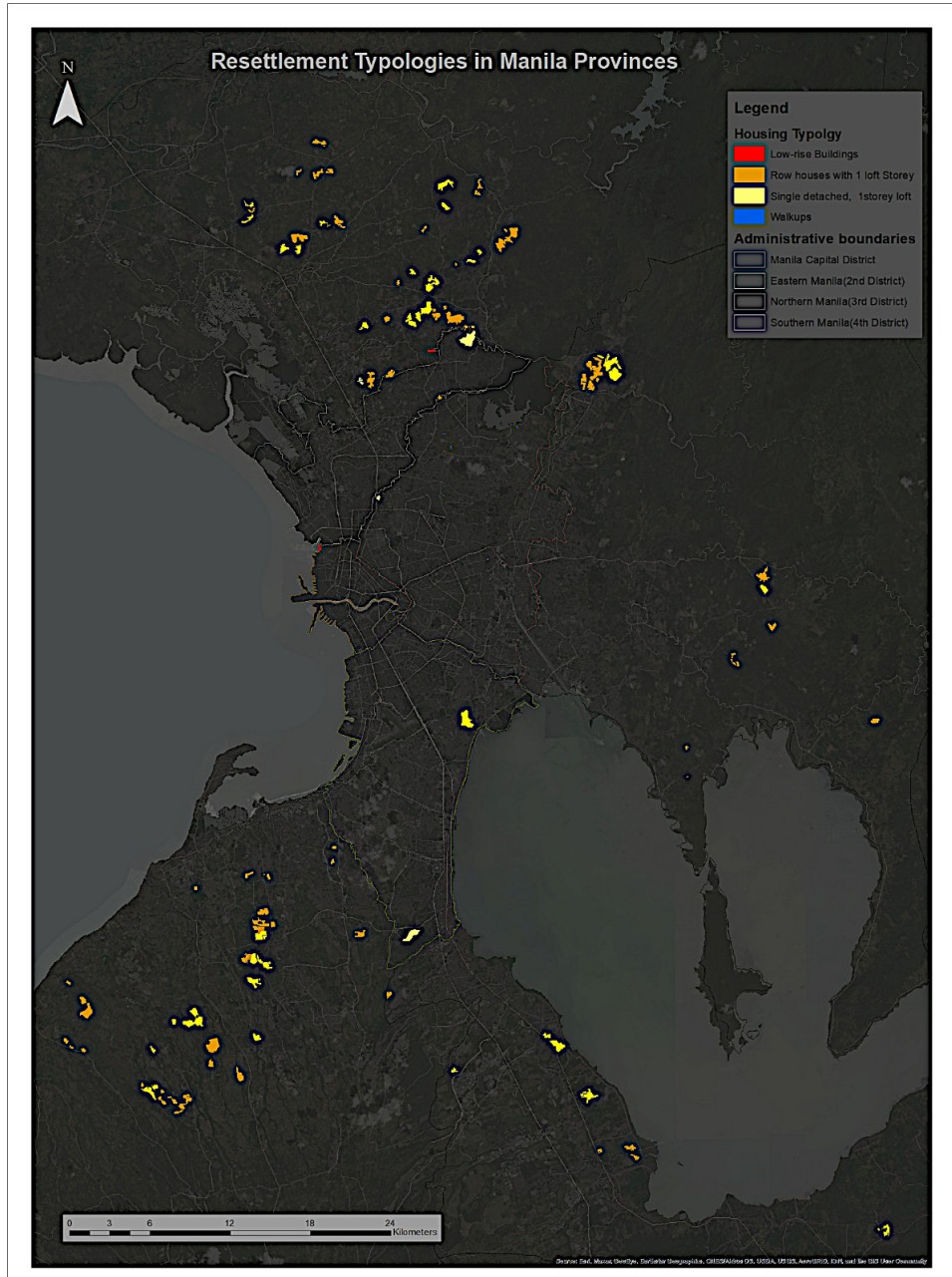

**Figure A3.** Resettlement projects selected regarding predominant housing types. Source: Project team design (Megha Kanaginahal) in 2020.

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
