# Peer review of "Managed Retreat as Adaptation Option: Investigating Different Resettlement Approaches and Their Impacts—Lessons from Metro Manila"

_sustainability, doi:10.3390/su13020829_

Round 1
Reviewer 1 Report
Major comments:
The article is very interesting and presents an important problem for years to come in some areas of developed countries. Specially, the managed retreat as a policy action for resolving climate change impacts.
Literature review is not well explained in the paper. A clear explanation about the managed retreat, the impacts and use are compulsory. Some literature is provided in the article but not well structured.
Concerning research approach and methods, point 2.1 gives us the categories of studies but, I insist, the review is not done. Point 2.2 explain that you did in-depth case analysis, “a project list of almost one hundred existing resettlement projects…. are developed” but a list (in annex or in a map) should be appreciated.
Results (point 3) are not results but an explanation of case study. I Think this is a case study description and, in my opinion, the point 4 Lessons Learnt are the results.
Work is done but a reorganization will improve the article.
Minor comments:
Line 167 expresses… although you introduces the reference at the end, is better to add Rutkowski (2015) expresses…..
To add some summary tables will help to understand better the information you provide as, for example, figure 3 that is very clear and help understand perfectly your explanation.
Line 425 and 426 The sentence needs better explanation: what’s the meaning of collect the amortization?
Line 504 and 505 “ This hypothesis is built on the insights derived from the large body of literature on failed resettlement projects and from results of the pre-workshop survey” but we don’t have the literature review about failed resettlements!
Line 543 and 544 “ However, the physical location is not the only decisive characteristic to influence the success of managed retreat. Other factors and resettlement components, such as the strategic approach of the involved stakeholders, the financing mechanisms in place, the housing typology or the participation possibilities also influence the outcome of projects”” What is based on? Is your opinion? If the factors are your contribution , these are very interesting but it is unclear if they are contribution or based on the literature.
Line 604 review: must be the to build
Line 644 Review The lastly mentioned change lastly to last
Author Response
Please see the attachment
Thank you very much for the comments and insightful suggestions!

Reviewer 2 Report
This paper presents a very interesting case of study with a problematic that deserves a careful methodological treatment to provide findings that can help to open discussions around future urban design actions and policies.
The literature review could include definitions of "managed retreat" displacement and relocation to avoid any misunderstanding.
If part of the methodology is literature review based the set of criteria to build the literature review needs to be presented and explained carefully to define the scope and boundaries of the research and the methods used to do it. It could be beneficial for the methodology to explain whether the literature review is topic based or problem based. If the it is problem-based the case studies used deserves a proper introduction and justification.
In line 126 what does “project-specific documents” mean? The section “in-depth case analysis” needs further development to define the subject of analysis and its assessment. The authors explained the intention behind the methodology, but the methods could be explained in detail to allow readers to redo the same experiment and get the same results.
The criteria for the qualitative analysis could benefited from the unfolding of the set of criteria used to do it. There is no explanation about the methods used in the workshop, the topics discussed neither the evidence of the information used. Maybe a clear and methodologic explanation of the workshops could be enough to write a concise and interesting paper.
Section 3.1 could benefit from the inclusion of concrete data, stats and numbers that provide a proper illustration of the situation. Try to use citations in such a way that the reader can have access to the specific information you are referring to.
In its present format it is not very clear what is the core of the paper. There are too many subjects. The discussion of all these topics at the same time are not providing an in-depth analysis of any of the topics referred. Having a research question and clear aims could contribute to frame the research. The relationship between the intention of poverty reduction and the lack of success of resettlements could be the core of the paper.
How were the “components” defined (line 306)?
The paper needs to include more maps to illustrate all the variables managed and to be more precise.
The paper can benefit from a data collection section where all the sources are named and mentioned.
The paper could use the analysis of specific case studies to establish a clearer comparative analysis. At the moment, this is very difficult to do.
The hypothesis derived from section 4 are difficult to link with the previous discussion since the hard-core data is not presented neither derived from a analysis section where specific case studies are used to show concrete results and evidence.
The mentioning of building resilience in section five (line 603) remains unexplained because it has not been treated in depth along the writing whose focus is a different topic. A link between resilience and the managed retreat of informal settlements could be the subject of a different paper.
Author Response
Please see the attachment.
Thank you very much for the comments and insightful suggestions!

Reviewer 3 Report
- the "urban poor" is often a category that includes people with lots of income differences. It can be hard to generalize when these differences exist.
- the limited response of the pre-workshop survey may not present enough data to support the conclusions.
Round 2
Reviewer 1 Report
I would like to start with appreciating the author’s responses.
I agree with the proposed order and now is more understandable. I would suggest to include in point 5 (Typology of Retreat ) the location: in Metro Manila.
You explained me that the literature review was done in a project named LIRLAP. I think a sentence explaining and linking to the project could improve this part.
I insist on the idea that the lessons learnt shouldn’t be considered as hypothesis. The main goal must be be to present your contribution. All of us have hypothesis that we should to demonstrate and, in case the contribution are lessons learned, this is completely acceptable.

Reviewer 2 Report
The new version of the article has improved its structure and legibility. The methodology sections is much clearer now and the appendix and figures are very helpful. The article could benefit from having only one research question instead of multiple questions and aims. Now, every single research question could be the subject of a new article. By having only one research question the paper can gain in depth and clarity. Since the literature review and discussion about urban resilience (in theory and practice) is not the core of the article neither it is developed in depth in any section of the paper, it could be beneficial to modify the title to avoid any confusion. Maybe "Investigating resettlement approaches and their impacts: lessons from Metro Manila" would be enough. Following this line, editing the topic of resilience out of the "ways forward" section could make it more coherent with the rest of the content.Author Response
Please see the attachment
